

# In search of traceability: two decades of calibrated Brewer UV measurements in Sodankylä and Jokioinen

J. S. Mäkelä[1,a], K. Lakkala[1,2], O. Meinander[1], J. Kaurola[1], T. Koskela[1],
J.M. Karhu[2], T. Karppinen[2], E. Kyrö[1], G. de Leeuw[1,3], and A. Heikkilä[1]

[1]Finnish Meteorological Institute, P.O. Box 503, 00101 Helsinki, Finland
[2]Finnish Meteorological Institute – Arctic Research Centre, Sodankylä, Finland
[3]University of Helsinki, Department of Physics, Helsinki, Finland
[a]now at: University of Jyväskylä, Jyväskylä, Finland

Received: 7 December 2015 – Accepted: 16 December 2015 – Published: 18 January 2016

Correspondence to: K. Lakkala (kaisa.lakkala@fmi.fi)

Published by Copernicus Publications on behalf of the European Geosciences Union.

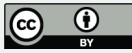

**GID**

doi:10.5194/gi-2015-40

**Traceability of Brewers**

J. S. Mäkelä et al.

**Abstract**

The two Brewer spectrometers of the Finnish Meteorological Institute at Jokioinen and Sodänkylä have been maintained and calibrated according to the highest levels of the WMO/GAW recommendations, with rigorous quality control and quality assurance.

However, the WMO/GAW recommendations are ambiguous on a number of decisions which need to be made when the response of the Brewer is calculated. The decisions have been left to the individual operators. Even within FMI, the two stations have used slightly different procedures, both completely consistent with the GAW recommendations. We suggest that the Brewer community should in the future address this ambiguity.

## 1  Introduction

The Brewer spectrophotometers are designed to measure the UV part of the solar spectrum. The absolute calibration of Brewer spectrophotometrers is a crucial part of the measurement chain needed to obtain accurate UV spectra. It is among the most difficult calibrations tasks in science, as there may be hundreds of channels that need to be considered; the signals at small wavelengths may be orders of magnitude smaller than at larger wavelengths; the uncertainties may differ at different wavelengths; the calibration lamps themselves may fade whenever they are used; and multiple transfer standards need to be used. Yet it is also a somewhat neglected part in the science literature. This is unfortunate, since as Garane et al. (2006) note "Achieving and maintaining a reliable absolute calibration of a UV spectroradiometer is a complicated process, but this is the most important requirement in UV spectroradiometry". Also, it is now almost certainly the largest unanalyzed source of uncertainty in the measurement chain.

Eleftheratos et al. (2014) note that "different studies have reported uncertainties between 5 and 7 % for global or direct spectral irradiance measurements in the UV-B,

Discussion Paper | Discussion Paper | Discussion Paper | Discussion Paper

**GID**

doi:10.5194/gi-2015-40

**Traceability of Brewers**

J. S. Mäkelä et al.

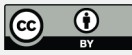

which are dominated by uncertainty in the calibration standards". This is in contrast to other sources of uncertainty, which have over the years been brought down significantly.

The existing WMO/GAW guidelines (Webb et al., 1998; Seckmeyer et al., 2001) are generic. They recommend that multiple lamps be used in each calibration, that calibrations be performed at least every six weeks, and that at least one lamp be calibrated against a laboratory-traceable lamp once a year. Many things are left open and up to the discretion of the operator: how the working lamps are chosen; when is a decision made that they need to be recalibrated; how exactly the daily response is to be calculated when new calibrations are made.

A review of the literature shows that each station does its own calibrations, and the methods and algorithms differ (Lam et al., 2002, 2007; Kazadzis et al., 2005; Garane et al., 2006; Lakkala et al., 2008; Simic et al., 2008; Schneider et al., 2008; Ialongo et al., 2008; Sabburg et al., 2001; Damiani et al., 2012). The FMI's Brewer measurements at Sodankylä and Jokioinen have followed the WMO recommendations to the highest realistic precision, with stability checks every three weeks and laboratory darkroom measurements every six. In this paper, we describe how the calibrations have been physically performed, as well as describe how the responsivity time series have been calculated over the years. We focus on the issues on which the GAW guideline offers little or no guidance, and which are rarely discussed in the scientific literature. The Brewer of Jokioinen has been moved to Helsinki in November 2015, and this paper thus also functions as a retrospective view of the Jokioinen Brewer measurements.

## 2 Data collection

### 2.1 Brewer spectrophotometers

The Brewer spectrophotometer (Brewer, 1973; Bais et al., 1996) is a standard device used to monitor UV radiation globally. The Finnish Meteorological Institute has

# GID

doi:10.5194/gi-2015-40

## Traceability of Brewers

J. S. Mäkelä et al.

operated Brewers at two locations, in Sodankylä (67.37° N, 27.63° E) and Jokioinen (60.82° N, 23.50° E). Both Brewers have been located at meteorological sounding stations which have had technicians present. The stations and the Brewers are described in more detail in a companion paper (Mäkelä et al., 2015).

## 2.2 Daily maintenance and stability checks

The devices are checked and cleaned regularly by the operators of the sounding stations. The Brewers are remotely operated, and daily quality and consistency checks are made autonomously. The internal mercury lamp is used to check the wavelength settings, while another internal lamp is used to monitor stability at the wavelengths used for ozone retrieval.

The stability and performance is monitored by regular 50 W lamp tests, using a portable calibrator of the type described in Kärhä et al. (2003). In practice, these are made approximately every three weeks, and they take about two hours. At any given time, there are about five different working 50W lamps of which three are used for any given calibration. This allows any drift to be monitored and caught quickly. The lamp measurements are usually done when the sun is below the horizon, so that measurements are not disturbed.

## 3 Calibration

### 3.1 Darkroom calibrations

Although measurements started in Sodankylä in 1990 and the measurements have been homogenized all the way back to that time, we focus on the years since 1998. At that point, both Jokioinen and Sodankylä started following the same calibration scheme which has been more or less in place since. Each station has one or two primary 1 kW standard lamps, whose absolute irradiance is measured in a National Standards Labo-

**GID**

doi:10.5194/gi-2015-40

**Traceability of Brewers**

J. S. Mäkelä et al.

Discussion Paper | Discussion Paper | Discussion Paper | Discussion Paper

ratory. Several 1 kW lamps are used as secondary standards. In addition, 50 W lamps are used to monitor the stability of the instrument. The primary standard lamps are changed as rarely as possible. The primary standard in either station has not changed since 2005.

⁵ The calibration of the UV measurements of the Brewer is performed using 1 kW DXW lamps in the dark room of the observatory. Calibrations are performed approximately every six weeks. In any given calibration, at least three lamps should be used. However, it is left open how these should be chosen from among the available calibration lamps. The lamps should be rotated, but the details are left to the discretion of the operator.

¹⁰ Care should be taken to burn the lamps different amounts of time. Lamps drift as they age, and at some point may need to be recalibrated or even rejected completely. There are no clear criteria to establish when this happens.

In Sodankylä currently, there are seven 1 kW lamps which are rotated. One of these is considered the primary standard, but for quality control purposes a second one is

¹⁵ also calibrated almost every year. In Jokioinen, only one primary lamp has been sent annually to the National Standards Laboratory. From the lamp set, excluding the primary standards, one is chosen to be measured each time, while the others are rotated. Figure 1 shows the raw output (counts(cycle)$^{-1}$ s$^{-1}$) from typical calibration runs in Jokioinen and Sodankylä. The differences between the Brewers are obvious: Jokioinen

²⁰ has a double monochromator and a much longer range; in addition, at 350 nm, the device switches the slit through which it measures the radiation.

Figures 2 and 3 show the measured counts at 305 nm for all lamps at both stations. Jokioinen has had two major discontinuities during which components were replaced, in June 2004 and December 2010. The data became discontinuous at those times,

²⁵ and this has to be considered in the homogenization. Sodankylä experienced no major discontinuities between 1998 and 2014.

**GID**

doi:10.5194/gi-2015-40

**Traceability of Brewers**

J. S. Mäkelä et al.

## 3.2 Calibration of the primary standards

Approximately once a year, the primary standards are calibrated at the National Standards Laboratory MIKES-Aalto. The annual laboratory calibration returns a so-called irx-file with the absolute irradiance spectrum of the primary standard at 0.5 nm intervals, with units of $\mathrm{mW\,m^{-2}\,nm^{-1}}$. Quite often, the Sodankylä primary lamps have also been measured in Jokioinen after each laboratory calibration. This cross-calibration has been an additional quality assurance, allowing cross-calibration of the lamps and the Brewers.

The absolute irradiances of three primary standards at 305 nm are shown in Fig. 4. It is seen that the lamps tend to fade as a function of time. The behaviour does not appear very linear, since the primary lamps may have been burned different numbers of times between laboratory calibrations. Plotting the irradiance as a function of number of burns (Fig. 5) shows that fading is close to linear. The best fit for this data is in fact exponential; the irradiance falls by approximately 0.3 % each time the lamp is burned. This demonstrates one of the major challenges in keeping a Brewer calibrated: every time the standard lamp is used in a calibration, it may change in an unpredictable and non-linear way. The fact that the primary standard fades so fast introduces a problem that is not mentioned in the GAW specification. In principle, the irradiance scale of the calibrated primary standard lamp should be immediately transferred to the secondary standard lamps. However, this would introduce a discontinuity in the results which does not physically exist.

## 4 Calculating spectral responsivity

### 4.1 Calculating irradiance files for the working lamps

FMI has developed simple QC software scripts to visualize and check the spectra, and estimate whether the spectrum of a given lamp has changed since the last calibration.

**GID**

doi:10.5194/gi-2015-40

**Traceability of Brewers**

J. S. Mäkelä et al.

Discussion Paper | Discussion Paper | Discussion Paper | Discussion Paper

At the moment, the decisions are made heuristically by the operator. Approximately a 1.0 % change is used as threshold to define whether the spectrum has changed. However, this is at best an approximate guideline, since the variation may be strongly wavelength dependent. The very smallest wavelengths are especially noisy, and thus individual channels may differ by considerably more than 1.0 %.

In practice, clear outliers (completely flawed measurements due to e.g. misalignment of a single lamp) can be seen very easily from the data, and rejected. By far the more problematic cases are those in which the changes are large for some wavelengths but small for others. There are no formal guidelines for these cases. In addition to the spectral measurements, the voltage of the lamp is monitored closely and used as a QC tool. Any fluctuations may be a sign that the lamp is becoming unreliable. These decisions are made heuristically by the oprator.

The exact calibration procedure at FMI has varied somewhat over the years, but a standard metrological philosophy is used: at any given moment, an irx-file is associated with every working standard. During each darkroom measurement, at least three standard lamps are used. If the operator judges that a given standard lamp is dimming or otherwise faulty, that standard lamp is recalibrated against the other lamps that were measured at the same time. The irx file of any given lamp may be recalculated at any time. This means that at any given time, the set of irx-files that is available is the most reliable one that can be obtained.

To ensure complete backward traceability, it is necessary to have an accurate record of the lamps that are valid for a given day, and the irx files that are associated with those lamps. One major question is how to take into account the discontinuity caused by the annual calibration of the primary standard. The absolute irradiance of a primary lamp may change by several percent between laboratory calibrations (depending on the wavelength). If this change is transferred directly to the secondary standards, there will be an unphysical discontinuity in the irx files.

# GID

doi:10.5194/gi-2015-40

## Traceability of Brewers

J. S. Mäkelä et al.

Interactive Discussion

## 4.2 Making the daily response time series

The formula for determining the true irradiance $I$ of an individual spectrum is:

$$I(\lambda) = C(\lambda)/R(\lambda),\tag{1}$$

where $C$ is the counts at a given wavelength $\lambda$ and $R$ is the response at that wavelength.
In practice, the calculation of the response is not standardized, and every Brewer operator does the calculation differently. Even within FMI, there have been minor differences between Jokioinen and Sodankylä.

In practice, the calculation of the response is a non-linear problem, because both the Brewer instrument and the calibration lamps tend to drift over time. The only absolute calibration of the primary lamp is performed once a year. The irradiance scale is then transferred to the other lamps by measuring them with the Brewer. By measuring the primary lamp, we get from Eq. (1) the spectral response of the Brewer on that specific day. Using this response to working lamp measurements of the same day, we can determine their irradiances. The irradiances of working lamps are then used to calculate the response time series of the Brewer during the other days of the year. In Fig. 6 we show an example of the counts measured by the Brewer during a lamp test. Figure 7 shows the corresponding .irx-file of the lamp and Fig. 8 shows the response of the Brewer calculated using Eq. (1).

For any individual lamp, if the number of counts changes between two calibrations, it is impossible to know whether the change is due to changes in the lamp or changes in the Brewer. For this reason, at least three lamps should be used in each calibration. When the lamps are cross-checked against each other, broken or severely drifting lamps can be identified and eliminated. However, there is no standard for determining how the averaging of the valid lamps should exactly be done to arrive at a response. Further, there is no standard way to determine the response for a specific day based on the calibrations that have been done at six-weeks intervals. Each Brewer station appears to be using its own solutions, which have very rarely been documented in detail in the open literature.

Discussion Paper | Discussion Paper | Discussion Paper | Discussion Paper | Discussion Paper |

**GID**

doi:10.5194/gi-2015-40

**Traceability of Brewers**

J. S. Mäkelä et al.

In Jokioinen, a linear time interpolation is performed separately for each lamp. Each wavelength is interpolated separately. The values for all lamps on any given day are then averaged. Finally, a running average of ±15 days is made for each wavelength to smooth out small-scale time variations. This is then the response for that day.

In Sodankylä, all the responses during a given calibration day are first averaged at each wavelength. As result there is just one measurement per each calibration day. Then a linear time interpolation is used to generate a response for each day of the year. These results are also smoothed with a runningaverage of ±15 days.

## 5 Discussion

We have started work on an analysis of the standard calibration lamp data. In particular, we are interested in seeing whether small changes in the calibration scheme will affect the final results. As shown by the difference between Jokioinen and Sodankylä, there are multiple possibilities which can be completely justifiable but which could produce somewhat different response time series. The subject of further studies will be to quantify the effect of using different pocedures to calculate the response time series.

The calibrations at FMI are fully consistent with the GAW/WMO specification. Yet, the two stations use different averaging schemes to arrive at a daily response. A literature study shows that every station appears to do this averaging differently, but still within the guideline. This suggests that there is considerable ambiguity in the standard. To some extent, such flexibility is an advantage, since it takes into consideration the fact that different stations may have vastly different resources and capabilities for carrying out the calibrations. There may even be stations for which only one annual calibration can be done. It may be completely impossible to define a scheme that would be universally usable.

Yet, we suggest that it would be beneficial for the Brewer community as a whole to come up with a more exact guideline for the calibrations. In terms of future develop-

**GID**

doi:10.5194/gi-2015-40

**Traceability of Brewers**

J. S. Mäkelä et al.

ment, a crucial step for the Brewer community would be the design of a better metadata management system for the lamp calibration data.

## 6 Conclusions

The two Brewers at FMI have been operated for about 20 years according to the highest quality guidelines defined by GAW/WMO. Nevertheless, the guidelines are loose enough that somewhat different absolute calibration schemes have been used at Jokioinen and Sodankylä, both justifiable both from the standards viewpoint and from the metrological viewpoint. We suggest that the Brewer community should attempt to make systematic recommendations for the calibration and transfer of irradiance standards. In the meantime, it is crucial that stations systematically document and archive the intermediate steps in the calibration. This will ensure true traceability to standards.

*Acknowledgements.* We thank the operators at Jokioinen and Sodankylä for the daily housekeeping of the Brewers, performing stability tests and the calibrations.

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

# GID

doi:10.5194/gi-2015-40

**Traceability of Brewers**

J. S. Mäkelä et al.

Discussion Paper | Discussion Paper | Discussion Paper | Discussion Paper | Discussion Paper

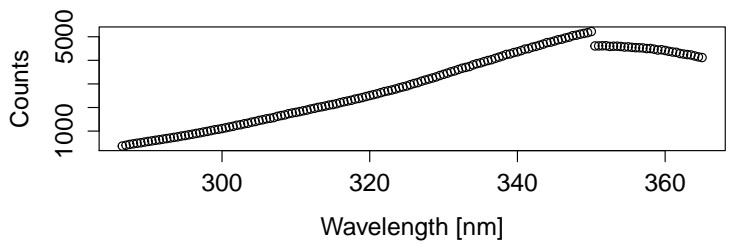

**Figure 1.** Typical raw outputs from a lamp calibration.

Discussion Paper | Discussion Paper | Discussion Paper | Discussion Paper | Discussion Paper

# GID

doi:10.5194/gi-2015-40

**Traceability of Brewers**

J. S. Mäkelä et al.

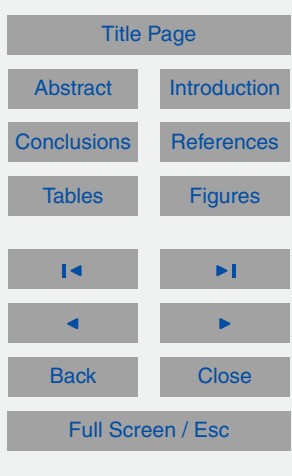

# GID

doi:10.5194/gi-2015-40

**Traceability of Brewers**

J. S. Mäkelä et al.

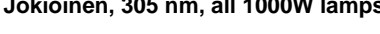

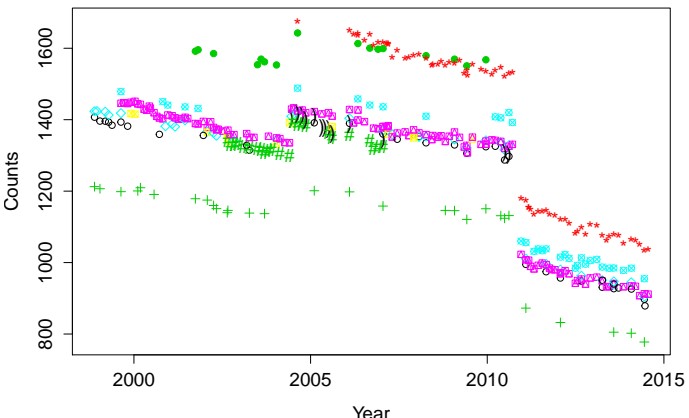

**Figure 2.** Raw counts from the calibrations of the primary and secondary 1000 W standard lamps in Jokioinen. Two discontinuities are clearly seen, during which the instrument characteristics changed significantly. In between these discontinuities, the counts fall steadily as the lamps age.

**GID**

doi:10.5194/gi-2015-40

**Traceability of Brewers**

J. S. Mäkelä et al.

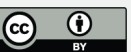

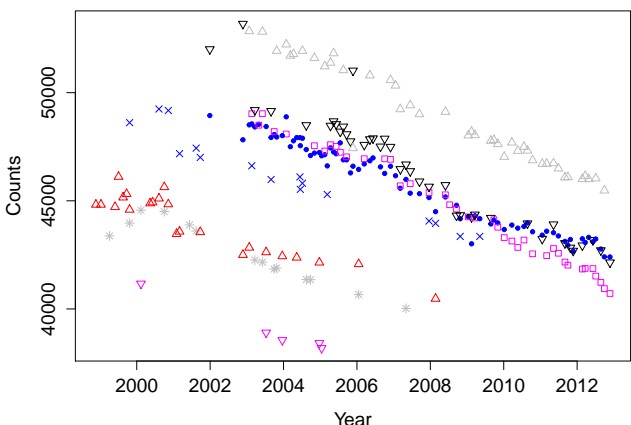

**Figure 3.** The raw counts from 1000 W standard lamp calibrations in Sodankylä up to 2013. There are no major discontinuities in the data.

# GID

doi:10.5194/gi-2015-40

**Traceability of Brewers**

J. S. Mäkelä et al.

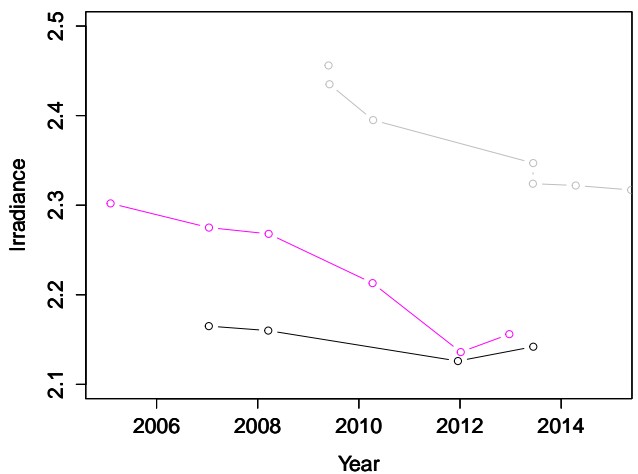

**Absolute irradiance at 305 nm, mW/m2/nm**

**Figure 4.** Time behavior of three primary standards that have been calibrated annually at the National Standards Laboratory MIKES-Aalto.

# GID

doi:10.5194/gi-2015-40

**Traceability of Brewers**

J. S. Mäkelä et al.

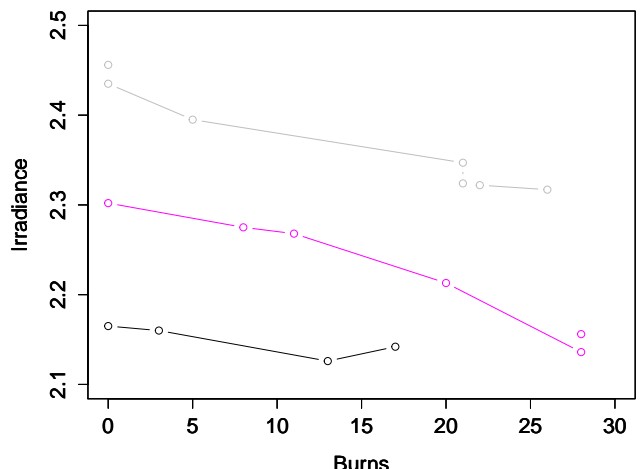

**Figure 5.** The data of Fig. 2, but plotted as a function of the number of times the lamp has been burned in darkroom calibrations. A linear fit to an exponential model suggests that the lamps fade by about 0.3 % every time they are burned.

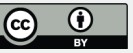

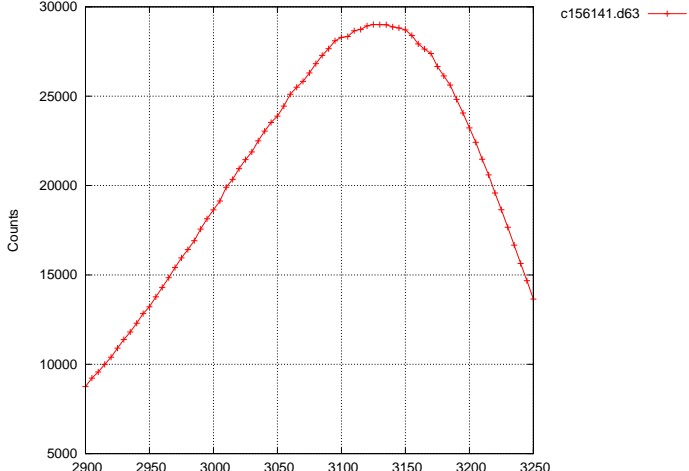

**Figure 6.** Example of a calibration. The raw spectrum for one lamp (d63) measured with the Brewer.

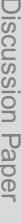

**GID**

doi:10.5194/gi-2015-40

**Traceability of Brewers**

J. S. Mäkelä et al.

**GID**

doi:10.5194/gi-2015-40

**Traceability of Brewers**

J. S. Mäkelä et al.

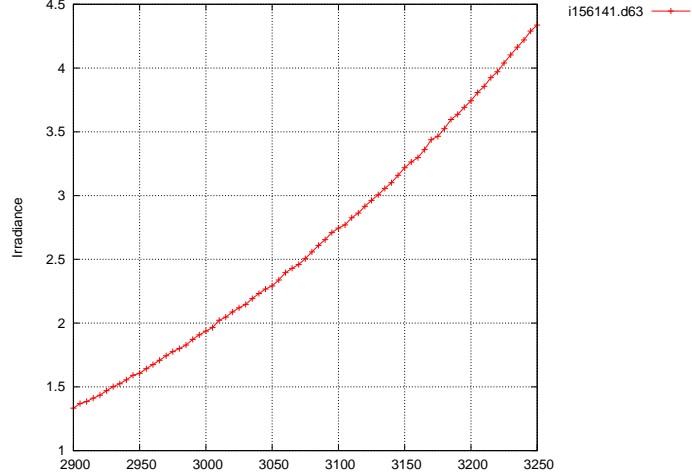

**Figure 7.** Example of a calibration. The irx values corresponding to the counts of Fig. 6.

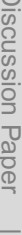

# GID

doi:10.5194/gi-2015-40

**Traceability of Brewers**

J. S. Mäkelä et al.

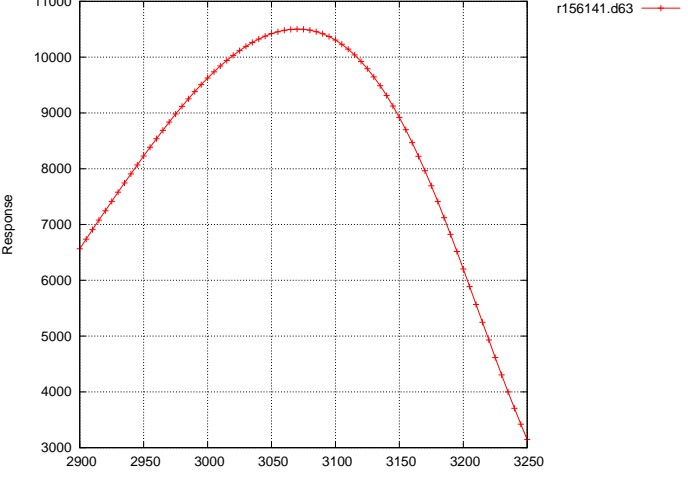

**Figure 8.** Example of a calibration. The response of the Brewer that was calculated from the measurement of the lamp d63 (Fig. 6) using the irradiance file showed in Fig. 7.