# Peer review of "In search of traceability: two decades of calibrated Brewer UV measurements in Sodankylä and Jokioinen"

_Geoscientific Instrumentation, Methods and Data Systems, 2015_

## Referee Comment (RC1) · Anonymous Referee #1 · 7 Feb 2016

The paper presents the calibration procedures followed at FMI for the calibration of the UV measurements conducted by two Brewer spectrophotometers. Such a study would be a useful contribution to the scientific community if it would provide the basis for assessing the quality of the data and the significance of the derived results in studies using and analyzing these data. From the title one would expect to see an assessment of the calibration history of the two Brewers and of course a discussion of the quality of the UV measurements. Unfortunately the paper does not provide any quantitative estimates that can be used to assess the uncertainly and quality of the data. It is a bare description of procedures and behavior of the lamps, but it does not get into the analysis of the measurements. The only worth publishing, quantitative result is

the estimation of the drift rate of the primary calibration lamps presented in section 3.2. My overall assessment is that the paper is not well written and needs substantial improvement before it is accepted for publication.

From the discussion in several places, I get the feeling that the paper criticizes the lack of standardized procedures for some steps of the calibration chain (no doubt that there are gaps), but I would expect proposing solutions on this issues.

There is need to polish the language, less in terms of the English, but mainly in terms of meaning and precision of the discussion. There are repetitions in different sections which confuse the reading of the paper.

In many places the discussion is very brief so that only experienced with the Brewer spectrophotometer readers can follow. For example, in line 21, page 5, the statement "the device switches the slit through which it measures the radiation" is not understandable by inexperienced Brewer users.

The figures are not very informative. More than half of them could be eliminated without affecting the discussion. The presentation quality can also be improved by adding legends to identify the different lines and symbols and by providing units of quantities in the axes titles.

The Discussion and Conclusions sections are very brief and really insufficient.

Specific comments

2, 23: "almost certainly the largest unanalyzed source of uncertainty". Indeed there is a lot of discussion in the scientific literature about the uncertainties in the absolute calibration. Please rephrase.

3, 25: I wouldn't say that the Brewer is "a standard device", because it may be regarded as "ideal". It could be probably better to say "a widely used instrument to monitor UV…."

4, 15: If lamp measurements are done only after sunset, doesn't this mean that they cannot detect any effects from temperature changes during daylight hours?

4, 24: Please state to which Standard Laboratory are the lamps traceable. It has been shown that there are differences among scales of different Laboratories.

5, 4-15: If the primary lamps are no replaced since 2005, how one can assure that they have not drifted during these 10-year period? From the discussion later on it seems that there are regular annual checks of the lamps calibration but this is not clear. In addition, please state if the seven lamps are secondary working standard lamps. Please clarify, distinguish the primary 1kW lamps that are calibrated at the National lab and the secondary 1 kW lamps which are usually calibrated locally.

5, 18: Is this really what it means? (counts(cycle)$-1$ s$-1$). Usually the output is in counts per second and not per cycle per second. Please check.

5, 22: These figures are the most important part of the results, because they can be directly related to the stability of the instrument's response. However, it would be much more valuable to draw the figures with the corrected signal after accounting for the drift in the calibration lamps, or better, the response function of the instrument. If the downward tendency of the signal is due to drifting of the lamps then the trend would be zero assuring that the instrument and eventually the measurements are of good quality. If not, then this has to be explained and discussed.

7, 25: The issue of applying the changes in the annual calibration of the lamps is very important, but the section ends without saying how they treat these changes. This is particularly important, if these changes are of the order of a few percent (as the authors state).

8, 15: The three figures 6-8 are not really necessary. They add nothing to the paper. Fig 6 is essentially the same as Figure 1b. In addition, there are no axes titles and units.

9, 1-8: The methods of applying backwards the calibration seems to be slightly different between the two stations. Is any of the two better or the final result is similar?

9, 17: This probably the most interesting part of the paper and I would really like to see a more extensive discussion and some clarifications on a few points. I understand form the text that these are the irradiance measurements reported by the Standards Laboratory each time the primary lamp has been sent for calibration.

If this is the case, why in the grey line (Fig. 4) there are two points at the same day in 2013? Why the last two points of the pink line in Fig. 4 are shown one on top of the other in Fig. 5?

I am not sure I see a very different behavior in the drifting of the lamps between Figures 4 and 5. Have you tried to plot the data against the burning hours of the lamps?

Technical

I suggest using consistently in the text "short" and "long" to define wavelength, instead of "small" and "large".

The caption of Figure 1: "Typical raw outputs from a lamp calibration" could better be "Typical spectral irradiance measurements of a calibration lamp as raw counts s-1".

---

## Short Comment (SC1) · 29 Feb 2016

In search of traceability: two decades of calibrated Brewer UV measurements in So-dankylä and Jokioinen, Mäkelä1, el al.

This is a well-written paper that will be of interest to the UV and ozone community. It clearly describes a detailed and professional approach to maintaining the calibration of Brewer spectrophotometers. As noted by the authors there is insufficient literature extant to guide the use of the instruments in making long-term UV spectral measurements. This paper is a very useful addition to that literature.

It is recommended that the paper be published with minor revisions as indicated.

[Figure]

Page 2, Line 14 '...difficult calibration tasks...' Singular

Page 3, Line 21 '...was moved...'

Page 8, Line 24 Suggest: '...exactly how the averaging of the valid lamps should be done...'

Page 8, Line 25 'Furthermore, ...'

Page 9 Not a comment on the paper: I found that the radiance of the lamp could be quite well modeled as a black body curve except for a local anomaly near 300 nm due to a tungsten absorption feature. This might provide a smoother way to interpolate radiances.

Page 9, Line 10. This referee feels that scientific literature should be written in the third person unless it is impossible to do so.

Page 9, Line 25 'Yet, it is suggested that...'

Page 10, Line 8 '...it is suggested that...'

Page 18. F6. Include the instrument number and type in the caption.

---

## Editor Comment (EC1) · N. Partamies (Editor) · 22 Mar 2016

The manuscript has received two good sets of comments from two independent reviewers. I would suggest the authors would provide their responses so that we can close the discussion.

---

## Author Comment (AC1) · 22 Jul 2016

Answers from the authors to the Interactive Comment on "In search of traceability: two decades of calibrated Brewer UVmeasurements in Sodankylä and Jokioinen" by J.S. Mäkelä et al.

Anonymous Referee #1

The authors wish to thank the Referee for his/her constructive criticism and invaluable suggestions for the improvement of the manuscript.

[Figure]

The comments are answered below in the following sequential manner: Q denoting the original comment; A denoting the authors' answer to the comment, and C denoting the corrections and amendments to the manuscript.

Q: The paper presents the calibration procedures followed at FMI for the calibration of the UV measurements conducted by two Brewer spectrophotometers. Such a study would be a useful contribution to the scientific community if it would provide the basis for assessing the quality of the data and the significance of the derived results in studies using and analyzing these data. From the title one would expect to see an assessment of the calibration history of the two Brewers and of course a discussion of the quality of the UV measurements. Unfortunately the paper does not provide any quantitative estimates that can be used to assess the uncertainly and quality of the data. It is a bare description of procedures and behavior of the lamps, but it does not get into the analysis of the measurements. The only worth publishing, quantitative result is the estimation of the drift rate of the primary calibration lamps presented in section 3.2. My overall assessment is that the paper is not well written and needs substantial improvement before it is accepted for publication.

A: We can see the point presented by the Referee. Our objective was to make a contribution by reporting on the procedures we follow in the maintenance of the irradiance scale of our spectrometers, including the regular recalibrations of the primary standard lamps, regular lamp measurements in our own optical laboratories at the home sites of our spectrometers, transfer of the irradiance scale into our secondary standard lamps, determination of the response of the instrument for the near-real time processing producing Level1 solar spectral UV irradiance data, and determination of the response time series for the post-processing producing Level2 data. We feel that these steps lay down the foundations for the traceability of the irradiance scale and the calibration of the spectrometers for solar spectral UV irradiance, and hence we hope that the title of the paper may be considered justified. At the same time, we realize that the intended scope of the paper should be expressed more clearly to the readers. We have therefore included a more precise description on the contents and objectives of the study in the abstract and in the introduction. In addition, we have carefully addressed the following guiding comments from the Referee and made the corresponding changes in the manuscript in an attempt to improve it.

Q1: From the discussion in several places, I get the feeling that the paper criticizes the lack of standardized procedures for some steps of the calibration chain (no doubt that there are gaps), but I would expect proposing solutions on this issues. There is need to polish the language, less in terms of the English, but mainly in terms of meaning and precision of the discussion. There are repetitions in different sections which confuse the reading of the paper. In many places the discussion is very brief so that only experienced with the Brewer spectrophotometer readers can follow. For example, in line 21, page 5, the statement "the device switches the slit through which it measures the radiation" is not understandable by inexperienced Brewer users.

A1: We agree with the Referee on the need of solutions to the areas with only loosely standardized procedures. We therefore present the procedures we have chosen to follow, and suggest joint pursuance towards common procedures. With the aid of the Referee's comment, we have examined the terminology used throughout the paper in an attempt to identify the places where changes should be done. We have also removed repetitive material and expanded discussion so that readers with no exhaustive prior knowledge on Brewer spectrophotometers can follow.

The change of slit is done at this wavelength to optimize the dynamic range of the detection of radiation and to avoid saturation of the signal at the same time. When the instrument is scanning solar UV irradiance, the change occurs exactly at the same wavelength. Therefore, it does not affect the calibration. Yet we would like to mention the change of the slit as it is the reason for the discontinuity in the raw counts of Brewer #107 shown in Fig. 1.

C1: The change in the slit is now mentioned and explained in Chapter 3 (Lamp measurements in on-site darkrooms) where we have added a more extensive description of the principles of the measurements. We also mention the change of the slit in the caption of Fig. 1 as the readers might wonder the discontinuity in the photon counts per cycle in the lamp scan made by the double monochromator.

Q2: The figures are not very informative. More than half of them could be eliminated without affecting the discussion. The presentation quality can also be improved by adding legends to identify the different lines and symbols and by providing units of quantities in the axes titles.

A2: We agree with the Referee on this point. We have removed the redundant figures. In the remaining figures, we have added the missing legends and axes titles. We have also added 3 additional figures to demonstrate the steps required in the derivation of the (Level1) responsivity used for near real-time processing and (Level2) responsivity time series used for post-processing of solar UV irradiance measurements.

C2: The following figures have been removed as redundant: Figs. 6-8.

In addition, we have made the following changes in the remaining figures 1-5: Figure1: Revised, two panels combined in one panel; unit in the y-axis corrected, counts in Sodankylä normalized to unity (counts per cycle) Figure 2: Revised, now as Fig. 3; legend added to distinguish the different lamps, unit in the y-axis corrected, wavelength now 311 nm Figure 3: Revised, now as Fig. 2; legend added to distinguish between the different lamps, unit in the y-axis corrected, wavelength now 311 nm; vertical lines added on days with abrupt jumps Figure 4: Revised; six primary standard lamps included, three for both stations, wavelength now 311 nm, irradiances from calibration performed on Bentham DTMc300 traceable to another irradiance scale now separated, error bars denoting the uncertainties given by the certificates included Figure 5: Revised; six primary standard lamps included, three for both stations, wavelength now 311 nm, error bars denoting the uncertainties given by the certificates included

We have also added the following figures 6-8: Figure 6. Responsivity of Brewer #037

as determined on the basis of the newly recalibrated primary standard lamps D22, D24, and D25 (discrete values) and the corresponding stepwise constant responsivities of type Level1 used for near real-time processing of solar UV irradiance spectra.

Figure 7. Responsivity of Brewer #107 as determined on the basis of the newly recalibrated primary standard lamps D01, D03, and D05 (discrete values) and the corresponding stepwise constant responsivities of type Level1 used for near real-time processing of solar UV irradiance spectra.

Figure 8. Demonstration of the phases in the determination of the final responsivity of type Level2 used for post-processing of solar spectral UV irradiance measured by Brewer #107 spectrophotometer.

The revised and new figures have been added as supplements to this reply.

All the figure captions has been added to the end of this reply.

Q3: The Discussion and Conclusions sections are very brief and really insufficient.

A3: We can see the need for a more comprehensive discussion and better organized conclusions. We have therefore substantially expanded the chapters "Discussion" and "Conclusions".

C3: We have revised the Discussion chapter as follows:

The original text was as follows:

"We have started work on an analysis of the standard calibration lamp data. In particular, we are interested in seeing whether small changes in the calibration scheme will affect the final results. As shown by the difference between Jokioinen and Sodankylä, there are multiple possibilities which can be completely justifiable but which could produce somewhat different response time series. The subject of further studies will be to quantify the effect of using different pocedures to calculate the response time series. The calibrations at FMI are fully consistent with the GAW/WMO specification. Yet, the

two stations use different averaging schemes to arrive at a daily response. A literature study shows that every station appears to do this averaging differently, but still within the guideline. This suggests that there is considerable ambiguity in the standard. To some extent, such flexibility is an advantage, since it takes into consideration the fact that different stations may have vastly different resources and capabilities for carrying out the calibrations. There may even be stations for which only one annual calibration can be done. It may be completely impossible to define a scheme that would be universally usable. Yet, we suggest that it would be beneficial for the Brewer community as a whole to come up with a more exact guideline for the calibrations. In terms of future development, a crucial step for the Brewer community would be the design of a better metadata management system for the lamp calibration data."

The text now reads:

"The lamp measurement data collected over the operational years of Brewer #037 and Brewer #107 spectrophotometer includes altogether 1931 and 2012 scans, respectively. The data allows retrospective examination of the changes occurred both in the lamps themselves and the responsivity of the instruments. Looking into the raw counts obtained from the lamp measurements of Brewer #037 and Brewer #107 already revealed long-term declination and abrupt jumps in the responsivity. The features observed in the raw counts may be considered truly attributable to changes in the responsivity as all the lamps indicate the same behaviour, independent from the frequency they are burned. However, separation of the fading of the lamps from the changes in the responsivity would be meaningful and should be possible by employing a linear mixed model, for instance. Regular recalibrations of the primary standard lamps enable identification of changes in the radiative output of the lamps. The temporal development of the primary standard lamps of Brewer #037 and Brewer #107 showed fading in general. Individual differences between the lamps obviously exist as regards their stability. The primary standard lamps used by Brewer #107 are burned more sparingly than the ones used by Brewer #037. This might be a feasible strategy

when aiming at minimizing the fading of the lamps. Following the temporal development of the radiative output of all the lamps is worthwhile since it also enables identification of the most stable lamps that would serve well as frequently scanned working standard lamps. The determination of the responsivity for both near real-time processing and post-processing of solar spectral UV irradiance includes phases where the operator has to make a choice between several alternatives. Up to the discretion of the operator are, for instance, which primary standard lamp/lamps is/are to be used as the basis of the determination of the responsivity of the instrument; which secondary/working standard lamps are to be used when a sudden change in the responsivity is obvious in between the recalibrations of the primary standard lamp and the Level1 responsivity for the near real-time processing needs to be updated; which lamps should be used when determining the Level2 responsivity for the post-processing of solar spectral UV irradiance data; how small variations in the responsivity would be meaningful to be retained both in Level1 and Level2 responsivity; how heavily the Level2 responsivity time series should be smoothed, etc. Quantified criteria would be very helpful for the operator making the decisions. The strategy of using multiple primary standard lamps that are regularly and rotationally recalibrated by NSL, and multiple secondary/working standard lamps that are frequently measured in the on-site laboratory, produces a lot of data that has to be regularly and carefully examined. They are, however, a prerequisite for identification of short-term changes in the responsivity of the spectrophotometer. The large amount of data that has to be processed with many scripts through many intermediate steps call for carefully designed data management. This is essential in an attempt to ensure the coherence in the continuance of the data processing."

In addition, we have rewritten Conclusions chapter as follows:

The original text was:

"The two Brewers at FMI have been operated for about 20 years according to the highest quality guidelines defined by GAW/WMO. Nevertheless, the guidelines are loose enough that somewhat different absolute calibration schemes have been used at

Jokioinen and Sodankylä, both justifiable both from the standards viewpoint and from the metrological viewpoint. We suggest that the Brewer community should attempt to make systematic recommendations for the calibration and transfer of irradiance standards. In the meantime, it is crucial that stations systematically document and archive the intermediate steps in the calibration. This will ensure true traceability to standards."

Conclusions now reads:

"Two Brewer spectrophotometers have been used to monitor solar spectral UV irradiance at Sodankylä and Jokioinen, Finland, for about 20 years according to the guidelines defined by GAW/WMO. The on-site measurements of 50 W and 1000 W lamps used for calibration and stability checks have produced a data set containing about 2000 scans for both instruments. The data enable retrospective examination of the temporal development of the response of the spectrometers. Both instruments show long-term decrease in their responsivities. In addition, abrupt changes as large as -25 % at 311 nm in the responsivity of Brewer #107 have occurred. The primary standard lamps regularly recalibrated by the National Standards Laboratory appear to fade at a rate of 0.05-0.6 % per burn. The finding encourages in sparing use of the primary standard lamps in the on-site laboratories. Examination of the responsivity time series based on three primary lamps revealed lags of up to 12 months in the detection of the abrupt changes. The lags were avoided and the sudden changes detected in due time by frequent measurements of the working standard lamps. This demonstrates the need for the frequent enough measurements of working standard lamps. The large amount of data accumulating from the lamp measurements and the multi-phase processing of the data calls for carefully designed data management. Future work should also include determination of quantified criteria to assist the operator in making the decisions in the various phases of the determination of the responsivity. This requires further research on the data, experimenting with the different choices and evaluating the consequences of each choice."

Specific comments Q4: 2, 23: "almost certainly the largest unanalyzed source of uncertainty". Indeed there is a lot of discussion in the scientific literature about the uncertainties in the absolute calibration. Please rephrase.

A4: We agree with the Referee on the abundance of the literature concerning this issue. We have expanded introductory discussion on the uncertainties related to the irradiance scales provided by the different National Standards laboratories, the transfer of the scale to the secondary standard lamps, and the standard lamp measurements carried out at home laboratories.

C4: We have revised the first paragraph in Introduction as follows. Originally, the text read:

"The Brewer spectrophotometers are designed to measure the UV part of the solar spectrum. The absolute calibration of Brewer spectrophotometrers is a crucial part of the measurement chain needed to obtain accurate UV spectra. It is among the most difficult calibrations tasks in science, as there may be hundreds of channels that need to be considered; the signals at small wavelengths may be orders of magnitude smaller than at larger wavelengths; the uncertainties may differ at different wavelengths; the calibration lamps themselves may fade whenever they are used; and multiple transfer standards need to be used. Yet it is also a somewhat neglected part in the science literature. This is unfortunate, since as Garane et al. (2006) note "Achieving and maintaining a reliable absolute calibration of a UV spectroradiometer is a complicated process, but this is the most important requirement 15 in UV spectroradiometry." Also, it is now almost certainly the largest unanalyzed source of uncertainty in the measurement chain. Eleftheratos et al. (2014) note that "different studies have reported uncertainties between 5 and 7% for global or direct spectral irradiance measurements in the UV-B, which are dominated by uncertainty in the calibration standards". This is in contrast to other sources of uncertainty, which have over the years been brought down significantly."

The paragraph now reads:

"The Brewer spectrophotometers are used to measure total atmospheric column ozone and UV part of the solar spectrum. The absolute calibration of a spectrophotometer is a crucial part of the measurement chain needed to obtain solar UV spectra with the lowest achievable uncertainty. In fact, maintenance of reliable absolute calibration may be considered as the most important requirement in UV spectroradiometry (Garane et al., 2006). The calibration is a challenging task due to several factors: the instrument uses not only one but a range of channels, one for each wavelength, that has to be considered; the signals at short wavelengths are orders of magnitude smaller than at longer wavelengths; the uncertainties differ at different wavelengths; the calibration lamps themselves fade whenever they are used; and multiple transfer standards need to be used. Estimations on the uncertainties in measurements of solar UV(-B) irradiance range from 5 to 7 %, the dominating source of uncertainty attributable to uncertainties in the calibration standards (Eleftheratos et al., 2014). Intercomparisons of the irradiance scales disseminated by the different National Measurement Institutes (NMIs) have varied from 2 to 5 % (Walker et al., 1991; Webb et al., 2003). While the uncertainties of the scales provided by NMIs are expected to decrease as the institutes re-establish their scales by linking them to primary detector scales, the uncertainties related to the performance of transfer standard lamps remain the key component in the overall uncertainty."

Q5: 3, 25: I wouldn't say that the Brewer is "a standard device", because it may be regarded as "ideal". It could be probably better to say "a widely used instrument to monitor UV: : :."

A5: We realize that the word "standard" may be interpreted as "ideal" so we agree with the Referee.

C5: We have changed the expression "standard device" to read "a widely used instrument to monitor solar UV radiation".

Q6: 4, 15: If lamp measurements are done only after sunset, doesn't this mean that

they cannot detect any effects from temperature changes during daylight hours?

A6: This would indeed be the case if we had no information on the temperature dependence of our Brewer spectrophotometers. However, we have characterized our Brewers for their temperature dependence in our optical laboratories in Jokioinen and Sodankylä. Each scan, be it a lamp or a sun scan, is corrected for the temperature dependence. In practice, the rule to make the measurements of the 50 W portable lamps after sunset is not strict, as the sun does not even go down at all in summer in Sodankylä. In other words, measurements of the portable lamps may be done outdoors in the evening before sunset. In that case, separate solar UV scans are taken in between the lamp scans. This ensures adequate sampling of the rest of the day for solar UV irradiance, enabling meaningful calculation of the daily UV doses.

C6: We have clarified the text as follows:

The original explanation read as follows: "The lamp measurements are usually done when the sun is below the horizon, so that measurements are not disturbed."

This is now rephrased as follows: -> "The outdoor lamp measurements are done in the evening, preferably after sunset. In case the measurements have to be made before sunset, separate solar UV scans are taken in between the lamp scans to ensure adequate sampling of the accumulating daily UV dose."

Q7: 4, 24: Please state to which Standard Laboratory are the lamps traceable. It has been shown that there are differences among scales of different Laboratories.

A7: We agree with the Referee that this is an essential piece of information. Over the years 1990-1998, calibrations for the primary standard lamps were ordered from four different laboratories: Gigahertz-Optik GmbH (GH), Germany; Optronics Laboratories Inc. (OL), US; STUK Radiation and Nuclear Safety Authority in Finland; and Statens Provningsanstalt (SP), Sweden. In 1999-2001 there was a period of transition during which the calibrations were ordered from SP, STUK, GH, and VTT MIKES Metrology

(formerly: MIKES Helsinki University of Technology HUT / MIKES Aalto University), the National Standards Laboratory (NSL) in Finland. Since Dec 2001, our primary standard lamps have been calibrated solely by NSL. The scale provided by NSL and currently used by the Brewer spectrophotometers #037 and #107 is traceable to Statens Provningsanstalt (SP) in Sweden.

The comparison measurements of spectral irradiance scales organized by the Consultative Committee for Photometry and Radiometry (CCPR) in 1990, 1996, and 2005 have indeed shown differences between the scales. Walker et al. (1991), for instance, have reported differences of 2-4 %. Differences detected in the irradiance scales were in fact a major reason for concentrating the recalibrations of the primary standard lamps of Brewer #037 and #107 exclusively into NSL in 2001.

Walker, J. H., Saunders, R. D., Jackson, J., & Mielenz, K. D.: Results of a CCPR intercomparison of spectral irradiance measurements by national laboratories, J. Res. Nat. Inst. Stand. Technol., 96, 647-668, 1991.

C7: We have added the following description of the suppliers of primary standard lamp calibrations in a new Chapter (4 Calibration of primary standard lamps):

"The calibration of the Brewer spectrophotometers is based on irradiance scale transferred to the on-site optical laboratories by primary standard lamps. Over the years 1990-1998, four different laboratories were used as suppliers of calibration of the primary standard lamps: Gigahertz-Optik GmbH (GH), Germany; Optronics Laboratories Inc. (OL), US; STUK Radiation and Nuclear Safety Authority in Finland; and Statens Provningsanstalt (SP), Sweden. The years 1999-2001 denoted a period of transition during which the calibrations were ordered from SP, STUK, GH, and VTT MIKES Metrology (formerly: MIKES Helsinki University of Technology HUT), the National Standards Laboratory (NSL) in Finland. Since Dec 2001, the calibrations have been ordered solely from NSL.

The comparison measurements of spectral irradiance scales organized by the Consultative Committee for Photometry and Radiometry (CCPR) in 1990, 1996, and 2005 have shown differences between the irradiance scales provided by different laboratories. Walker et al. (1991) reported differences of 2-4 % in the 1990 intercomparison in the UV wavelengths. A spread of $\pm 5$ % at wavelength 300 nm was obtained in the 1996 intercomparison (Webb et al., 2003). Concentrating the recalibrations of the primary standard lamps of Brewer #037 and #107 exclusively into NSL has removed the uncertainty related to the differences in the irradiance scales.

The primary standard lamps are 1-kW tungsten-filament incandescent halogen lamps of type DXW operated in vertical orientation in a distance of 50 cm of the focal plane of the diffuser. The calibration of the primary standard lamps at NSL is carried out by using the method for the realisation of the detector-based spectral irradiance scale (Kübarsepp et al. 2000). The absolute responsivity of the used trap detector is traceable to the cryogenic electrical substitution radiometer of Statens Provningsanstalt SP, Sweden. In the 1990's, lamps manufactured by GH and OL were used. Currently, all the primary standard lamps regularly used in Jokioinen and Sodankylä are manufactured by GH."

Q8: 5, 4-15: If the primary lamps are no replaced since 2005, how one can assure that they have not drifted during these 10-year period? From the discussion later on it seems that there are regular annual checks of the lamps calibration but this is not clear. In addition, please state if the seven lamps are secondary working standard lamps. Please clarify, distinguish the primary 1kW lamps that are calibrated at the National lab and the secondary 1 kW lamps which are usually calibrated locally.

A8: We agree that this issue has not been clearly enough dealt with in the manuscript. The primary lamps are recalibrated by the National Standards Laboratory VTT MIKES Metrology every 1-2 years. The primary lamps have been measured in our own laboratory 1-3 times per year in Jokioinen and 2-4 times per year in Sodankylä. The lamp may be expected to fade in some extent every time the lamp is burned. It is therefore likely that some drift has occurred during the 10-year period. Our examination on the

NSL certified irradiances at 311 nm revealed overall drifts ranging from -0.4 % to -7.6 %, translating into annual drifts ranging from -0.03 % to -0.81 %.

We have added a list of the standard lamps currently used as primary standards. We have also added explanation on the roles of primary standard, secondary standard, and working standard lamps.

C8: We have added Table 1 (attached as a pdf supplement to this reply) listing all the primary standard lamps currently and regularly used in the laboratory measurements of Brewer #037 and Brewer #107.

In addition, we have added the following paragraph in Chapter 2 (now titled "Brewer spectrophotometer and its calibration for solar spectral UV irradiance measurements) explaining the use of primary and secondary standards:

"The calibration of the Brewer spectrophotometers #037 and #107 is based on primary (reference) standard lamps regularly recalibrated in a National Standards accredited laboratory VTT MIKES Metrology (hereinafter denoted as NSL) and regular lamp measurements in the on-site optical laboratories in Sodankylä and Jokioinen. Measurements of 1000 W standard lamps are used as a basis for the determination of the responsivity of the instruments. In addition to the primary standard lamps recalibrated regularly by NSL, several secondary and working standard lamps are used. The irradiance scale is transferred from the primary standard lamps to the secondary and working standard lamps, to avoid unnecessary burning and consequent premature fading of the primary standards. Secondary standard lamps are used to preserve the calibration provided NSL transfered to them and therefore burned sparingly. Working standard lamps are used most frequently in the on-site laboratory measurements."

Q9: 5, 18: Is this really what it means? (counts(cycle)-1 s-1). Usually the output is in counts per second and not per cycle per second. Please check.

A9: The Referee is right on this point. The unit should not be counts cycle-1 s-1.

However, it should not be counts per second either, but counts cycle-1 or just counts.

Figure 1 shows examples of dark count corrected raw counts produced and printed out into files named XLjjjyy.107 and ULjjjyy.037 (jjj denoting the Julian date, yy denoting the year) for the lamp scans performed by Brewer #107 and Brewer #037, respectively. The raw counts are collected and recorded by the spectrometers as described in the following.

The photons entering the spectrometer are selectively passed onto the detector (cathode of the photomultiplier tube). During the scan, the dispersing grating(s) are rotated by step motors to pass on photons with nominally only one wavelength at a time. Six separate exit slits are positioned after the (first) grating. A slotted rotating slit mask is used, driven by another step motor, to expose only one or two of the slits at a time, or to prevent radiation to enter none of the slits, to enable measurement of the dark counts. In Brewer #037, the slit mask is positioned at the exit slit of the monochromator. In Brewer #107, the slit mask is positioned at the entrance of the second (recombining) monochromator. The slit mask cycles back and forth between optical endstops.

In external lamp scans of Brewer #107, the slitmask is cycled through 30 oscillations for wavelengths shorter than 300 nm and through 20 oscillations for wavelengths longer than 300 nm. The photon counts $F_i$ are accumulated through slit #1 for wavelengths shorter than 350 nm and through slit #5 for wavelengths longer than 350 nm. In addition, the dark counts $F_1$ from one of the slit mask positions is recorded. Brewer #037 uses 30 cycles over its whole wavelength range (290-325 nm). In Brewer #107, the dark current corrected photon counts ($F_i – F_1$) are output (in XLjjjdd.107) as normalised to 1 cycle observations. In Brewer #037, they are output (in ULjjjdd.037) as such, accumulated over the total of 30 cycles.

The next step of processing converts the dark current corrected raw counts ($F_i – F_1$) into units counts s-1. The PMT has a prescaler that divides the photon pulses by 4. The actual counts (number of photons) recorded is therefore $4 \times (F_i – F_1)$. The time taken
by CY slit mask cycles back and forth is 2xCYxIT, with IT denoting the integration time pre-defined as 0.1146 s. Hence, the count rate (in units counts s-1) may be derived using the formula $C_i=(4 \cdot (F_i-F1))/(2 \cdot CY \cdot IT)=(2 \cdot (F_i-F1))/(CY \cdot IT)$. From this point on, it is the count rate that is further processed. However, the raw counts stored into the files XLjjjyy.107 and ULjjjyy.037 are indeed counts, not counts per second.

To plot the raw counts stored in lamp scan files XLjjjyy.107 and ULjjjyy.037, two alternative ways exist. We could plot the readings as such. In that case, the unit would be counts. Alternatively, we could present the data as the number of counts collected per each cycle. Then, the unit would be counts cycle-1. As the stored counts are normalized to 1 for Brewer #107 but not normalized for Brewer #037, we would like to use the latter way, i.e.: to present the raw counts shown in Figure 1 as counts cycle-1. The signal in the lower panel of Figure 1 (raw counts recorded by Brewer #037) should be therefore divided by factor 30. This has been changed in the revised manuscript.

C9: We have added a description on how the photons are collected by Brewer spectrophotometers, to ensure that also readers not familiar with the instrument get an idea. The description included in Chapter 3 (titled "Lamp measurements in on-site darkrooms") reads as follows:

"The photons entering the spectrometer are selectively passed onto the detector, the cathode of the photomultiplier tube. During the scan, the dispersing grating(s) are rotated by step motors to pass on photons with nominally only one wavelength at a time. Six separate exit slits are positioned after the (first) grating. A slotted rotating slit mask is used, driven by another step motor, to expose only one or two of the slits at a time, or to prevent radiation to enter none of the slits, to enable measurement of the dark counts. In Brewer #037, the slit mask is positioned at the exit slit of the monochromator. In Brewer #107, the slit mask is positioned at the entrance of the second (recombining) monochromator. The slit mask cycles back and forth between optical end stops.

In external lamp scans of Brewer #107, the slitmask is cycled through 30 oscillations for wavelengths shorter than 300 nm and through 20 oscillations for wavelengths longer than 300 nm. The photon counts Fi are accumulated through slit #1 for wavelengths shorter than 350 nm and through slit #5 for wavelengths longer than 350 nm. This can be seen as a jump in the curve of the raw counts of Brewer #107 in Fig. 1. In addition, the dark counts F1 from one of the slit mask positions is recorded. Brewer #037 uses 30 cycles over its whole wavelength range (290-325 nm). In Brewer #107, the dark current corrected photon counts (Fi – F1) are output (in XLjjjdd.107) as normalised to 1 cycle observations. In Brewer #037, they are output (in ULjjjdd.037) as such, accumulated over the total of 30 cycles."

We have also revised Fig. 1 in a way that it now gives the counts normalized to one measurement cycle for both instruments. The unit in the y-axis has been corrected to read "counts cy-1".

Q10: 5, 22: These figures are the most important part of the results, because they can be directly related to the stability of the instrument's response. However, it would be much more valuable to draw the figures with the corrected signal after accounting for the drift in the calibration lamps, or better, the response function of the instrument. If the downward tendency of the signal is due to drifting of the lamps then the trend would be zero assuring that the instrument and eventually the measurements are of good quality. If not, then this has to be explained and discussed.

A10: We agree and we see the need for a more detailed examination and expansion of the discussion on this point. We have revised Figs. 2 and 3. They now include legends listing the different lamps. The unit is counts cycle-1 as explained above. The effects of the changes in the instrument's response and the fading of the lamp are indeed superimposed in the recorded counts per cycle. Yet the figures may provide some preliminary information on the changes in the responsivity, since the counts recorded from all the lamps follow the same pattern, no matter how frequently they are burned. It therefore appears that the burning time of the lamp has notably smaller effect than the

long-term change in the instrument's responsivity. The general tendency is downwards so the responsivity is degrading. For Brewer #107, abrupt changes in the responsivity also occur. Short-term changes may be also seen in counts of Brewer #037, although in a smaller scale. We have added explanation on these points into the manuscript.

Separation of the fading of the lamps from the changes in the responsivity is somewhat complicated by the abrupt changes especially in the readings of #107, but should be possible by employing a linear mixed model, for instance. This is something we aim to look into in a continuing study. However, to respond to the suggestion on examining the temporal development of the corrected signal or the response of the instruments in more detail, we have prepared a description on the determination of the Level1 responsivity used for near-real-time processing of data for both instruments with accompanying figures. The responsivity is based on measurements of the primary standard lamps taken in the on-site optical laboratories in Jokioinen and Sodankylä immediately after a recalibration in NSL. From the discrete points of the determined responsivities, the stepwise constant Level1 responsivity time series are formed. They provide an insight into the true changes in the responses of the instruments, cleaned from the effect of the fading of the lamps.

C10: We have added the following explanation on the potential of the raw counts in revealing changes in the responsivity of the instrument:

"The time series of the raw counts may be examined as such to examine the temporal development of the responsivity of the instrument. Figures 2 and 3 shows plots of raw counts at 311 nm extracted from the measurements of a selection of 1000 W lamps. The wavelength of 311 nm was chosen to be used throughout this study, since this wavelength has been measured also in the earlier scans in the 1990's. The effects of the changes in the instrument's response and the fading of the lamp are superimposed in the recorded counts per cycle. Yet the figures may provide some preliminary information on the changes in the responsivity, since the counts recorded from all the lamps follow the same pattern, no matter how frequently they are burned. It therefore appears

that the fading of the lamp has notably smaller effect than the long-term change in the instrument's responsivity. The general tendency in the response of both instruments is declining. In Brewer #107, three abrupt changes in the responsivity are obvious: the first in Jun 2004, the second in Jan 2010, and the third in Aug 2014. Short-term changes may be also seen in raw lamp counts of Brewer #037, although in a smaller scale."

We have also included a description on the determination of the Level1 and Level2 responsivities as new chapters, and prepared new figures (Figs. 6-8) to illustrate the procedure and the outcome. The chapters read as follows:

"4.1 Level1 responsivity for near real-time processing

The responsivity of the spectrophotometer is determined for the day the (primary standard) lamp is measured in the on-site laboratory. Usually, at least three different 1000 W standard lamps are measured during the same day. After a recalibration of the primary standard lamps in NSL, several secondary and working standard lamps are measured in a row during a laboratory session of 1-2 days, to transfer the irradiance scale from the newly recalibrated primary standard lamps to the secondary and working standard lamps. The responsivity may be based on one single lamp or on an average of 2-3 trusted lamps. The obtained responsivity is assumed to stay constant until the next scan of a standard lamp indicating a change in the responsivity of the instrument.

Figures 6 and 7 show the responsivities of Brewer #037 and #107 determined on the basis of measurements of three primary standard lamps in the on-site laboratories immediately after the recalibrations of the lamps in NSL From the discrete points of the determined responsivities, the stepwise constant (Level1) responsivity time series are formed. These kinds of responsivities are used in the near real-time processing of the solar UV measurements made by Brewer spectrophotometers #037 and #107. The operator may choose to fix the responsivity onto one lamp only or use an average of

2-3 lamps.

The stepwise Level1 responsivities shown by Figs. 6 and 7 give insight into the true changes in the responses of the instruments, cleaned from the effect of the fading of the lamps. The responsivity of Brewer #037 seems to decline fairly steadily. The largest drop of the order of -5 % is detected in the measurements of lamps D24 and D25 since the beginning of 2014. The abrupt changes obvious in the raw counts (Fig. 3) are also seen in the responsivity of Brewer #107. During 2002-2012, the responsivity at 311 nm has declined approx. by 4 %. The largest drop in the responsivity is seen in the beginning of 2012. The change is approx. -25 %. It is noteworthy that the change took place already in Dec 2010. In case no measurements of working standard lamps had been taken between the relatively sparse measurements of primary standard lamps, the Level1 responsivity used for the near real-time processing of solar UV irradiance spectra would have remained onto a faulty level for more than 12 months.

4.2 Level2 responsivity for post-processing In addition to the near real-time processing of solar UV irradiance measurements made by Brewer #037 and #107 spectrophotometers, producing Level1 data, the scans are retrospectively post-processed to produce Level2 data. This allows the operator to account for even the small scale variations in the responsivity that were neglected in the near real-time processing. In addition, it allows the operator to retrospectively view the behaviour of each individual lamp, separate the true changes in the instrument from the changes in the lamp, and choose the trusted lamps to serve as the basis of the determination of the responsivity for each period of time.

To demonstrate the determination of the Level2 responsivity, the lamp measurements collected in 2015 are used and each phase in the process is described in the following. The phases are illustrated in Fig. 8.

Phase 1 The primary standard lamps D01 and D05 were recalibrated in NSL on May 19, 2015. They were measured by Brewer #107 in the on-site darkroom in Jokioinen on

Jun 23, 2015 (Julian date 174/15). D01 and D05 have both proved very stable in time as may be also seen in Figs. 4 and 5. The determination of the Level2 responsivity is decided to be based on these two primary standard lamps. The responsivity for the instrument for day 174/15 is defined as the average of the responsivity obtained from the measured count rates and newly certified irradiances of D01 and D05. This will serve as an anchor point for the daily responsivity to be determined for the selected time period. The irradiance scale may be now transferred to all the lamps measured on-site on day 174/15 using the formula (2). This is done here for the frequently used but very stable and hence trusted working standard lamps D14 and D42. In practice, certificates for the irradiance of lamps D14 and D42 on day 174/15 are produced, using the responsivity determined on the basis of the measurements of lamps D01 and D05. The responsivity of the instrument is further calculated on the basis of the count rates obtained from the measurements of D14 and D42 on day 174/15. The procedure positions the responsivities determined on the basis of the lamps D14 and D42 both in the middle of those obtained on the basis of the lamps D01 and D05 on day 174/15.

During 2015, lamps D01 and D05 have been measured in the on-site laboratory also on Jan 14, 2015 (014/15) and Nov 25, 2015 (329/15). Irradiance of the lamps on those days is assumed the same as on day 174/15, i.e.: the same as the readings in the certificate given by NSL dated May 19, 2015. The working standard lamps D14 and D42 have been also measured on days 013/15 and 329/15. They can be therefore calibrated against the average of D01 and D05 on these two additional measurement days in the same way as on day 174/15. Again, the transfer of calibration positions the responsivities determined for lamps D14 and D42 in the middle of those determined for the primary standard lamps D01 and D05. The working standard lamps D14 and D42 have been measured by Brewer #107 in the on-site laboratory on days 056/15, 098/15, 223/15 and 272/15 in addition to the days 013/15, 174/15, and 329/15. These measurements provide four additional points in time to be used in the determination of the responsivity time series. For the time period 174/15-329/15 this is fortunate, as there is a short-term change in the responsivity of the instrument detected by the lamps D14

and D42. Measurements with the 50 W lamps during the same time period indicate the same change, so it may be concluded that the change is real. The irradiances for the lamps D14 and D42 are calculated by linear interpolation between the irradiances fixed onto the scale provided by the lamps D01 and D05 for days 013/15, 174/15, and 329/15. The corresponding responsivities for the lamps D14 and D42 are calculated from the measured count rates and the assumed irradiance. Finally, an average of the responsivities determined for the lamps D14 and D42 is calculated, marked as black crosses in Fig. 8.

Phase 2 The discrete responsivities derived as an average of the lamps D14 and D42, calibrated against the average of the primary standard lamps D01 and D05, are next used to derive a time series of responsivity. The discrete points are connected with linear interpolation in time, resulting in a time series in a form of a polyline. The obtained time series, giving daily responsivity for Brewer #107, could be already used as such in post-processing of solar UV irradiance data. The time series is plotted with a thick grey line in Fig. 8.

Phase 3 The polyline shaped time series of responsivity derived in Phase 2 contain sharp turning points. In most cases it may be assumed that in reality the changes are not that sudden in the responsivity of the instrument. The time series may be therefore filtered using a moving average with a window of a suitable width. This is done here using a window of width of 31 days. The selection of the width depends on how small variations in the time series are to be retained and in which extent the sharp turns in the time series are to be smoothed away. The resulting time series is plotted with a thin black line in Fig. 8. This is considered a Level2 responsivity time series that could be used for the post-processing of the solar spectral UV irradiance measurements collected during the year 2015.

The demonstration presented here is confined into the lamp measurement data collected during 2015. The responses derived on the basis of the measurements of the lamps D01 and D05 in the on-site laboratory on day 014/15 appear to be fairly far from

each other. It is possible that irradiance of either one of the lamps has changed as compared to that determined by NSL on day 174/15. Alternatively, the count rates recorded during the scan of either one of the lamps may be faulty as a result of instabilities in the lamp voltage, for instance. In practice, the previous recalibrations of the primary standard lamps are also taken into account, as well as the Level2 responsivity derived in the previous post-processing event. This ensures that the reference points in the time series of the responsivity are fixed onto correctly chosen trusted lamps."

Q11: 7, 25: The issue of applying the changes in the annual calibration of the lamps is very important, but the section ends without saying how they treat these changes. This is particularly important, if these changes are of the order of a few percent (as the authors state).

A11: We see the point made by the Referee. We have added descriptions on the determination of Level1 responsivity used for near real-time processing of data and Level2 responsivity used for post-processing of data.

C11: We have added new chapters titled "4.1 Level1 responsivity for near real-time processing" and "4.2 Level2 responsivity for post-processing", given in C10 above.

Q12: 8, 15: The three figures 6-8 are not really necessary. They add nothing to the paper. Fig 6 is essentially the same as Figure 1b. In addition, there are no axes titles and units.

A12: We see the need to revise the Figs. 6-8 in a way that they will provide an insight into the determination of the responsivity of the instruments. We have now added description on the steps required and revised the Figs. 6-8 accordingly.

C12: We have replaced Fig. 6-8 with new figures describing the steps taken in the determination of the Level1 and Level2 responsivities of the instruments.

Q13: 9, 1-8: The methods of applying backwards the calibration seems to be slightly different between the two stations. Is any of the two better or the final result is similar?

A13: We realize the need for a more extensive explanation on the procedure used for deriving the daily responsivity. The responsivity of the instruments is indeed based on the primary standard lamps recalibrated regularly in VTT MIKES Metrology, the laboratory with the National Standards accreditation in Finland. This fixes the irradiance scale to that maintained by Statens Provningsanstalt SP in Sweden. The scale is transferred to the secondary standard and working standard lamps. This is realized by measuring the irradiance of the newly NSL recalibrated primary standard lamps and the secondary/working standard lamps consequtively during a laboratory session of 1-2 days in the dark rooms of Jokioinen and Sodankylä. The responsivity used in the near real-time processing of data (Level 1 responsivity) is updated with the responsivity based on the measurements of the newly recalibrated primary standard lamps. The final responsivity time series used in post-processing of data (Level 2 responsivity) is based on measurements of the primary standard and/or secondary standard lamps regularly taken in the on-site laboratories. In Sodankylä, the primary standard lamps are more frequently measured, so they can be used as such as a basis for the determination of the Level 2 responsivity. In Jokioinen, the primary standard lamps are measured more sparingly in the on-site laboratory, to minimize the fading of the lamps, and the working standard lamps are measured more frequently instead. In this case, the determination of the responsivity time series is based on working standard lamps that have been calibrated against the primary standard lamps, measured in the on-site laboratory frequently enough, and found stable enough, both electrically and in terms of their radiative output.

The two averaging schemes mentioned on page 9, line 17, refer to the two alternative ways to determine the Level 2 responsivity in case it is based on measurements of more than just one lamp. The two different ways are included as alternative options in the processing script. Measurements of lamp irradiances in the on-site laboratory produce discrete points of responsivity. The Level 1 responsivity is derived from these discrete points obtained for the primary standard lamps as a stepwise constant time series. For the final Level 2 responsivity time series, the discrete points are interpolated

in time, resulting in continuous polylines. The software allows one to choose whether the discrete responsivities obtained for the measurements of individual lamps are to be first averaged and then interpolated, or if the interpolation is to be performed first for the discrete responsivities obtained for each and every lamp and after that averaged. The intermediate results are different (one time series vs. time series for each and every lamp), but the final result is the same. The choice is left to the operator of the instrument and is mainly based on his/her preferences. Examination of the responsivity time series for each and every lamp separately may perhaps more easily reveal any loss of coherence between the lamps.

We have added full description on the procedures used to derive the Level 1 and Level 2 responsivities. The step-by-step description is illustrated by new figures. We hope that the readers would find the description and the figures informative.

C13: We have added full description on the procedures used to derive the Level1 and Level2 responsivities. The step-by-step description is illustrated by new figures 6-8.

Q14: 9, 17: This probably the most interesting part of the paper and I would really like to see a more extensive discussion and some clarifications on a few points. I understand form the text that these are the irradiance measurements reported by the Standards Laboratory each time the primary lamp has been sent for calibration. If this is the case, why in the grey line (Fig. 4) there are two points at the same day in 2013? Why the last two points of the pink line in Fig. 4 are shown one on top of the other in Fig. 5? I am not sure I see a very different behavior in the drifting of the lamps between Figures 4 and 5. Have you tried to plot the data against the burning hours of the lamps?

A14: The question concerning the two averaging schemes mentioned on page 9, line 17, have been dealt with in the answer to the previous comment. Concerning the Referee's comment on Figs. 4 and 5: The two points in the grey line at the same day in 2013 correspond to the calibration measurements carried out using different methods. Most of the NSL recalibrations of our primary standard lamps are performed using the

method for the realisation of the detector-based spectral irradiance scale (Kübarsepp et al. 2000). The absolute responsivity of the used trap detector is traceable to the cryogenic electrical substitution radiometer of Statens Provningsanstalt SP, Sweden. These recalibrations have provided us the irradiance certificates we have used for the calibration of the Brewer spectrophotometers. However, some of the primary standard lamps, like D22, D24 and D25, are also recalibrated by NSL in an extended wavelength range of 250-2100 nm or 250-2500 nm. These calibrations are performed using Bentham DTMc300 and a scale traceable to MRI (Metrology Research Institute, Finland) for wavelengths below 900 nm, and to NPL (National Physical Laboratory, UK) for wavelengths above 900 nm. The resulting certificates are used for other radiometers with wavelength range extending to the infra-red but not used in the determination of the responsivities of Brewer spectrometers. However, we would prefer keeping the points in Fig. 2 as they illustrate a difference between two irradiance scales. The difference is less than 1 % at 311 nm. We have now separated in Fig. 2 the certified irradiances determined with the Bentham measurements and traceable to another irradiance scale from the ones used for the calibration of the Brewer spectrometer #037. We have included the irradiances from the certificates received for all the lamps currently in use as primary standard lamps (3 for Brewer #037 and 3 for Brewer #107). The plot is now for 311 nm instead of the previous 305 nm. The wavelength 311 nm was chosen to be used throughout the study because the scanning scheme used by Brewer #037 in the 1990's used the step of 35 Å over 2900-3250 Å, so the wavelength 3050 Å is not included in those scans. We have also added the uncertainties provided by the NSL into Figs. 4 and 5 as error bars, to illustrate the stability and the true changes in the stability of the irradiance of the lamps. We have added explanative text into the figure caption and into the chapter describing the role of the primary standard lamps.

The reason why the pink and the blue line have departed from each other in Fig. 5 in comparison to Fig. 5 is the difference in the x-axis. In Fig. 4, the certified irradiance at 305 nm is plotted as a function of time. In Fig. 5, it is plotted as a function of burns. The time the lamps are burned in every lamp measurement event is not strictly constant,

but the operators try to restrict the burning to the minimum. Presenting the data of Fig. 5 as a function of burning hours would therefore appear almost identical to the current presentation as a function of burns. We would wish to retain Fig. 5 since it describes how the lamps fade as they are burned. On the basis of Fig. 5 it is easier to conclude that the stability of the lamps D01, D03 and D05 is at least partly due to more sparing use.

Kübarsepp, T. , Kärhä, P., Manoocheri, F., Nevas, S., Ylianttila, L., and Ikonen, E.: Spectral irradiance measurements of tungsten lamps with filter radiometers in the spectral range 290 nm to 900 nm, Metrologia 37, 305-312, 2000.

C14: We have explained the occurrence of the two points slightly apart from each other in Fig. 4. The text included as a paragraph in Chapter 3 (titled "Calibration of primary standard lamps") is as follows:

"The filled circles in Fig. 4 denote parallel calibrations performed using Bentham DTMc300 spectroradiometer, extending over wavelengths 250-2100 nm or 250-2500 nm. The scale is traceable to MRI (Metrology Research Institute, Finland) for wavelengths below 900 nm, and to NPL (National Physical Laboratory, UK) for wavelengths above 900 nm. These certificates are not used for the calibration of Brewer spectrophotometers, but for other radiometers with wavelength range extending to the infra-red. However, they illustrate a difference between two irradiance scales. The difference is less than 1 % at 311 nm."

Technical Q15: I suggest using consistently in the text "short" and "long" to define wavelength, instead of "small" and "large".

A15: We agree with the Referee that this is indeed conventional way to express the matter. We have followed the suggestion. As a result, adjectives "short" and "long" are used throughout the text instead of "small" and "large".

C15: We have replaced the word "small" (page 2, line 16; page 7, line 4) and the word

"large" (page 2, line 17) with the words "short" and "long".

Q16: The caption of Figure 1: "Typical raw outputs from a lamp calibration" could better be "Typical spectral irradiance measurements of a calibration lamp as raw counts s-1".

A16: We realize that this is a more precise expression and hence we have followed the suggestion.

C16: The Figure caption of Fig.1 has been rephrased according to the suggestion of the Referee. The original formulation was: "Typical raw outputs from a lamp calibration." The caption now reads: "Typical spectral irradiance measurements of a calibration lamp as raw counts cycle-1. Brewer #037 measurement taken with primary standard lamp D24 on Mar 20, 2012. Brewer #107 measurement taken with primary standard lamp D01 on Jan 26, 2012."

Figure captions of the revised and new figures:

Figure 1. Typical spectral irradiance measurements of a 1000W standard lamp as raw counts per cycle. Brewer #037 measurement taken with primary standard lamp D24 on Mar 20, 2012. Brewer #107 measurement taken with primary standard lamp D01 on Jan 26, 2012.

Figure 2. Time series of the raw counts (in units counts per cycle) at 311 nm recorded by Brewer #037 spectrophotometer in measurements of a selection of 1000 W lamps. The lamps marked with an asterisk are primary standard lamps regularly recalibrated in VTT MIKES Metrology.

Figure 3. Time series of the raw counts (in units counts per cycle) at 311 nm recorded by Brewer #107 spectrophotometer in measurements of a selection of 1000 W lamps. The lamps marked with an asterisk are primary standard lamps regularly recalibrated in VTT MIKES Metrology.

Figure 4. Irradiance at 311 nm as certified by the NSL of the primary standard lamps used in calibration of Brewer #037 in Sodankylä (D22, D24, and D25) and Brewer

**107 in Jokioinen (D01, D03, and D05). The filled circles denote parallel calibrations performed using Bentham DTMc300, extending over wavelengths 250-2100 nm or 250-2500 nm, traceable to another irradiance scale.**

Figure 5. Fading rate of the primary standard lamps used in calibration of Brewer #037 in Sodankylä and Brewer #107 in Jokioinen, given by the NSL certified irradiance at 311 nm as a function of the number of burning events.

Figure 6. Responsivity of Brewer #037 as determined on the basis of the newly recalibrated primary standard lamps D22, D24, and D25 (discrete values) and the corresponding stepwise constant responsivities of type Level1 used for near real-time processing of solar UV irradiance spectra.

Figure 7. Responsivity of Brewer #107 as determined on the basis of the newly recalibrated primary standard lamps D01, D03, and D05 (discrete values) and the corresponding stepwise constant responsivities of type Level1 used for near real-time processing of solar UV irradiance spectra.

Figure 8. Demonstration of the phases in the determination of the final responsivity of type Level2 used for post-processing of solar spectral UV irradiance measured by Brewer #107 spectrophotometer. The dashed lines are plotted to guide the eye.

Please also note the supplement to this comment:
http://www.geosci-instrum-method-data-syst-discuss.net/gi-2015-40/gi-2015-40-AC1-supplement.pdf

**Fig. 1.**

Fig. 2.

Legend:
- D22*
- D24*
- D25*
- D62*
- D08*
- D63
- D61

Fig. 3.

Fig. 4.

[Figure]

**Fig. 5.**

Chart: Certified irradiance @311 nm / mW nm⁻¹ m⁻² vs Burns, with series D01, D03, D05, D22, D24, D25.

[Figure]

**Fig. 6.**

Figure: Plot of Responsivity @311nm / counts s⁻¹ W⁻¹ m² nm versus Year, with legend:
- D22 based (pointwise)
- D24 based (pointwise)
- D25 based (pointwise)
- D22 Level1 (stepwise)
- D24 Level1 (stepwise)
- D25 Level1 (stepwise)

**Fig. 7.**

[Figure]

**Fig. 8.**

**Supplement:**

Table 1. Information on the primary standard lamps used in calibration of Brewer #037 and #107.

| Lamp | Primary to | 1$^{st}$ calibration | Last calibration | Number of (re)calibrations | Scans at home laboratory |
|------|-----------|---------------------|------------------|---------------------------|-------------------------|
| D01 | Brewer #107 | 26.3.2003 | 19.5.2015 | 9 | 44 |
| D03 | Brewer #107 | 14.12.2001 | 29.4.2014 | 4 | 42 |
| D05 | Brewer #107 | 20.8.1999 | 19.5.2015 | 7 | 44 |
| D31 | Brewer #107 | 26.3.2003 | 31.1.2005 | 2 | 46 |
| D41 | Brewer #107 | 20.4.2010 | 21.12.2011 | 2 | 16 |
| D24 | Brewer #037 | 3.2.2005 | 21.5.2015 | 10 | 59 |
| D22 | Brewer #037 | 14.12.2001 | 6.1.2012 | 8 | 51 |
| D25 | Brewer #037 | 14.1.2007 | 14.6.2013 | 5 | 56 |
| D62 | Brewer #037 | 14.6.2013 | 21.5.2015 | 4 | 10 |
| D08 | Brewer #037 | 11.1.2004 | 11.1.2004 | 1 | 18 |

---

## Author Comment (AC2) · 22 Jul 2016

Answers from the authors to the Interactive Comment on "In search of traceability: two decades of calibrated Brewer UV measurements in Sodankylä and Jokioinen" by J.S. Mäkelä et al.

Referee #2

The authors wish to thank the Referee for his comments and suggestions for the revision of the manuscript.

[Figure]

The comments are answered below in the following sequential manner: Q denoting the original comment; A denoting the authors' answer to the comment, and C denoting the corrections and amendments to the manuscript.

Q: This is a well-written paper that will be of interest to the UV and ozone community. It clearly describes a detailed and professional approach to maintaining the calibration of Brewer spectrophotometers. As noted by the authors there is insufficient literature extant to guide the use of the instruments in making long-term UV spectral measurements. This paper is a very useful addition to that literature. It is recommended that the paper be published with minor revisions as indicated.

Q1: Page 2, Line 14 '...difficult calibration tasks...' Singular

A1: We agree. The expression is now corrected.

C1: We have replaced the original expression '...difficult calibrations tasks...' with 'a challenging task'

Q2: Page 3, Line 21 '...was moved...'

A2: We agree. The sentence has been corrected.

C2: We have corrected and rephrased the sentence to read "In November 2015, Brewer #107 was moved to Helsinki (60.20° N, 24.96° E) where it has been operated ever since." In the revised manuscript, the sentence appears at the end of the first paragraph of Chapter 2 now titled "2 Brewer spectrophotometer and its calibration for solar spectral UV irradiance measurements".

Q3: Page 8, Line 24 Suggest: '...exactly how the averaging of the valid lamps should be done...'

A3: The sentence has been removed in the revised manuscript, as a part of a chapter rewritten completely.

Q4: Page 8, Line 25 'Furthermore, ...'

[Figure]

A4: This particular sentence has been removed in the revised manuscript, as a part of a chapter rewritten completely. However, we have used the connector word "Furthermore" in another place in the text.

Q5: Page 9 Not a comment on the paper: I found that the radiance of the lamp could be quite well modeled as a black body curve except for a local anomaly near 300 nm due to a tungsten absorption feature. This might provide a smoother way to interpolate radiances.

A5: We wish to thank the Referee on this point.

Q6: Page 9, Line 10. This referee feels that scientific literature should be written in the third person unless it is impossible to do so.

A6: We realize that the use of the third person is indeed often recommended to be used in scientific literature. We have checked the manuscript and revised all occurrences of the first person pronoun "we".

C6: Nine occurrences of the first person pronoun "we" have been removed in the revised manuscript and replaced by an expression using the third person.

Q7: Page 9, Line 25 'Yet, it is suggested that...'

A7: We agree.

C7: We have removed the first person pronoun "we" and rephrased the sentence using the third person.

Q8: Page 10, Line 8 '...it is suggested that...'

A8: We agree.

C8: We have removed the first person pronoun "we" and rephrased the sentence using the third person.

Q9: Page 18. F6. Include the instrument number and type in the caption.

A9: We agree.

C9: We have revised all the figures, including Fig. 6. The figures now include the information necessary for the identification of the instruments and the individual lamps.

---

## Author Comment (AC3) · 22 Jul 2016

We have carefully addressed all the questions and comments presented by the Referees and submitted our response.
* * *